# Prediction of type 2 diabetes risk in people with non-diabetic hyperglycaemia: model derivation and validation using UK primary care data

Briana Coles [ID],[1,2] Kamlesh Khunti [ID],[1,2] Sarah Booth,[3] Francesco Zaccardi,[1,2] Melanie J Davies,[2] Laura J Gray[3]

[1]Leicester Real World Evidence Unit, Diabetes Research Centre, University of Leicester, Leicester, UK
[2]Diabetes Research Centre, University of Leicester, Leicester General Hospital, Leicester, UK
[3]Department of Health Sciences, University of Leicester, Leicester, UK

**Correspondence to**
Ms Briana Coles;
bc188@leicester.ac.uk

## ABSTRACT

**Objective** Using primary care data, develop and validate sex-specific prognostic models that estimate the 10-year risk of people with non-diabetic hyperglycaemia developing type 2 diabetes.

**Design** Retrospective cohort study.

**Setting** Primary care.

**Participants** 154 705 adult patients with non-diabetic hyperglycaemia.

**Primary outcome** Development of type 2 diabetes.

**Methods** This study used data routinely collected in UK primary care from general practices contributing to the Clinical Practice Research Datalink. Patients were split into development (n=109 077) and validation datasets (n=45 628). Potential predictor variables, including demographic and lifestyle factors, medical and family history, prescribed medications and clinical measures, were included in survival models following the imputation of missing data. Measures of calibration at 10 years and discrimination were determined using the validation dataset.

**Results** In the development dataset, 9332 patients developed type 2 diabetes during 293 238 person-years of follow-up (31.8 (95% CI 31.2 to 32.5) per 1000 person-years). In the validation dataset, 3783 patients developed type 2 diabetes during 115 113 person-years of follow-up (32.9 (95% CI 31.8 to 33.9) per 1000 person-years). The final prognostic models comprised 14 and 16 predictor variables for males and females, respectively. Both models had good calibration and high levels of discrimination. The performance statistics for the male model were: Harrell's C statistic of 0.700 in the development and 0.701 in the validation dataset, with a calibration slope of 0.974 (95% CI 0.905 to 1.042) in the validation dataset. For the female model, Harrell's C statistics were 0.720 and 0.718, respectively, while the calibration slope was 0.994 (95% CI 0.931 to 1.057) in the validation dataset.

**Conclusion** These models could be used in primary care to identify those with non-diabetic hyperglycaemia most at risk of developing type 2 diabetes for targeted referral to the National Health Service Diabetes Prevention Programme.

## INTRODUCTION

People with blood glucose levels raised beyond normal but not high enough for

### Strengths and limitations of this study

► A large, representative primary care database was used to develop the models using HbA1c to quantify blood glucose.
► A range of predictors were considered specifically selected due to clinical relevance to development of type 2 diabetes.
► The cohort was split into development and validation datasets instead of using a fully external database to validate the model, but given the size of the cohort and the large number of events, this likely had little effect on model development.
► The outcome for this study was defined using a single medcode or test result indicating type 2 diabetes.

a formal diagnosis of type 2 diabetes (ie, glycated haemoglobin [HbA1c] 6.0%–6.4% or 42–47 mmol/mol) are at high risk of eventually developing type 2 diabetes. This high-risk state has been termed non-diabetic hyperglycaemia (NDH) or pre-diabetes.[1] In 2015, in England, it was estimated that there were five million people aged 16 years and over with NDH, a prevalence of 11.4%.[1] The prevalence was much lower in people younger than 40 years of age, with the exception of minority ethnic populations.[1] Evidence from large-scale clinical trials has shown that the development of type 2 diabetes can be delayed or even prevented if those with NDH are enrolled into a diabetes prevention programme (DPP).[2 3]

DPP encourage participants to change their behaviour with a focus on increasing physical activity, improving diet quality and reducing weight. These programmes have been developed and tested internationally.[2 4–6] Initially studies focused on very intensive programmes—for example, a programme developed and tested within the USA involved 16 one-to-one individualised

sessions over 6 months, followed by monthly individual and group based sessions to reinforce messages.[4] Over a mean follow-up of 2.8 years, there was a 58% reduction (95% CI 48% to 66%) in the risk of type 2 diabetes in those randomised to receive the prevention programme compared with standard care.[4] Other studies conducted in Finland and China with similar programmes found comparable results.[5 6] Such resource intensive programmes, although very effective, are not viable for delivery within an National Health Service (NHS) setting.

Therefore, emphasis shifted to developing a more pragmatic programme that could be delivered in a group setting and requires less contact time. The NHS DPP launched in 2016 and is open to adults with NDH.[7 8] The NHS estimates that once the NHS DPP is fully rolled out in 2020, 100 000 people will access the programme each year.[9] Based on this, it will take over 50 years for all those with NDH to access the programme.

Many prognostic and diagnostic models have been developed and validated for identifying those with undiagnosed type 2 diabetes, NDH or those at risk of developing type 2 diabetes.[10–12] Evidence shows that the risk of developing type 2 diabetes in those with NDH is variable. Some people with NDH will revert to normal glucose levels over time, with only a subset going on to develop type 2 diabetes.[13] Indeed referring all patients with NDH to the DPP is overtreatment in the majority of cases.[14] Therefore, in the era of big data and personalised medicine, using data stored in primary care to target referrals to those at highest risk may be a more efficient use of the NHS DPP than the current blanket referral approach.

To date, no validated risk assessments for use in those with NDH have been developed for use in the UK. Therefore, we developed and validated sex-specific prognostic models to quantify the 10-year risk of those with NDH developing type 2 diabetes using data routinely collected in primary care. Such models should be used to target referrals to the NHS DPP.

## METHODS
### Study design and data source
This observational retrospective cohort study included a sample of primary care patients from the UK who were registered with practices contributing to the Clinical Practice Research Datalink (CPRD). The CPRD includes anonymised primary care electronic health records for over 11.3 million patients from 674 UK practices dating back to 1987.[15] The CPRD includes data for approximately 6.9% of the UK population and is broadly representative of the age, sex and ethnicity of the UK general population.[15] When available, patients were also linked to Office of National Statistics (ONS) to obtain the date of death and Hospital Episode Statistics (HES) to obtain ethnicity (both available for 59% of patients in the study cohort). Linked Index of Multiple Deprivation data (quintiles) were also obtained.

This study included an open cohort of patients registered in CPRD aged 18 years or older with NDH. NDH was defined as an HbA1c measure within 42–47 mmol/mol (6.0%–6.4%). For each patient, the index date was defined as the first recorded test measurement indicating NDH between 1 January 2000 and 31 December 2017. Patients with a diagnosis of type 2 or type 1 diabetes before the index date were excluded. Patients with an HbA1c measure greater than 47 mmol/mol (6.4%), random blood glucose measure greater than 11.0 mmol/L (199 mg/dL), or fasting plasma glucose measure greater than 6.9 mmol/L before the index date were also excluded as these patients were assumed to be in the process of confirming a diagnosis of type 2 diabetes. Patients prescribed metformin, the current first line therapy for type 2 diabetes, were also excluded. Patients were followed up for a maximum of 10 years until diagnosis of type 2 diabetes, or censoring (transferring out of practice, death or the end of study on 31 December 2017, whichever came first).

The cohort was split into a development and validation dataset. To split the cohort, practices of registration were stratified by region and patients were clustered by practice (see online supplemental table 1). Approximately 33% of practices in each region were randomly assigned to the validation dataset.

### Sample size
There were 71 063 males and 83 642 females meeting the inclusion criteria (see online supplemental figure 1). This resulted in 50 049 males and 59 028 females in the development dataset and 21 014 males and 24 614 females in the validation dataset. Within the development dataset, 4719 males and 4613 females developed type 2 diabetes. Riley et al[16] have proposed an approach for calculating the minimum number of events per predictor parameter for a survival model based on the model's anticipated R squared, event rate, follow-up time and number of predictor parameters. We used the R squared, event rate and mean follow-up for men and women from a similar study to estimate the required sample size.[17] For women, based on 31 predictor parameters (deprivation has five categories) considered for our study, the required minimum sample size was 3406. For men, based on 29 predictor parameters considered for our study, the required minimum sample size was 2585.

### Outcome
The outcome was the first diagnosis of type 2 diabetes recorded within the CPRD between 1 January 2 000 and 31 December 2017. The first diagnosis of type 2 diabetes was identified by medcode; HbA1c measure greater than 47 mmol/mol (6.4%); random blood glucose measure greater than 11.0 mmol/L (199 mg/dL); or fasting plasma glucose measure greater than 6.9 mmol/L.

**Table 1** Potential predictor variables

| Demographic information | |
| --- | --- |
| Age | Ethnicity |
| Sex | Deprivation |
| **Medical/family history** | |
| Family history of diabetes | Polycystic ovary syndrome |
| Cardiovascular disease | Sleep apnoea |
| Schizophrenia or bipolar affective disorder | Depression |
| Learning disabilities | Renal/kidney disease |
| Gestational diabetes | |
| **Prescribed medications** | |
| Antihypertensives | Statins |
| Corticosteroids | Aspirin |
| Second generation 'atypical' antipsychotics | |
| **Clinical measurements** | |
| HbA1c | Pulse rate |
| Body mass index | Serum cholesterol |
| Systolic blood pressure | Liver function test |
| Diastolic blood pressure | Waist circumference |
| **Lifestyle factors** | |
| Smoking status | Alcohol use |

## Predictor variables

We examined potential predictor variables based on established risk factors for type 2 diabetes and those risk factors included in existing risk scores for type 2 diabetes-related outcomes.[10–12 17 18] Table 1 shows the predictor variables considered.

Data on demographic factors, medical and family history, prescribed medications, clinical measurements and lifestyle factors were obtained from CPRD (and HES for ethnicity). Age in single years at the index date was used. Ethnicity was derived from HES as white or non-white and when unavailable, the most recent code in CPRD was used. Deprivation was measured using the 2010 Index of Multiple Deprivation quintiles (1=least material deprivation; 5=most material deprivation). The closest value to the index date was selected for continuous measures including body mass index (BMI), systolic and diastolic blood pressure, pulse rate, serum cholesterol, liver function test and waist circumference, restricting to values recorded within 6 months before the index date. BMI is automatically calculated within the medical record based on input height and weight. Biologically implausible values were excluded including serum cholesterol outside of 1–15 mmol/L, systolic blood pressure outside of 20–250 mm Hg, diastolic blood pressure outside of 30–150 mm Hg and BMI outside of 9–96 kg/m². Prescribed medications (yes or no) were determined from one or more prescription records within 6 months before the index date. Alcohol use (entity type=5) and

smoking (entity type=4) were defined using records indicating current smoking or alcohol use within 1 year before the index date. All others were considered non-current smokers and/or alcohol users—including former smokers and/or alcohol users. Medical and family history was determined from a diagnosis code before the index date.

## Handling of missing data

Potential predictor variables with missing data for more than 33.3% of the study cohort were excluded, as these are most likely not collected as part of routine primary care (see online supplemental table 2). Assuming data were missing at random and based on previous research, multiple imputation was used to generate five imputed datasets.[17 19] Missing ethnicity (white or non-white), serum cholesterol and systolic and diastolic blood pressure were imputed using chained equations.

## Development of the models

Modelling was performed using the Stata stpm2 command for fitting flexible parametric survival models on the log cumulative hazard scale.[20] Null flexible parametric models were fitted to estimate type 2 diabetes risk using between 1 and 5 df to model the baseline hazard function: the final df was determined from visual examination of the plots of the baseline hazard functions as well as Akaike information criterion (AIC) and Bayesian information criterion (BIC) statistics. Multivariable fractional polynomial models were considered that included fractional polynomial transformations of potential continuous predictor variables. This process selects fractional polynomial models that best predict the outcome of interest. Then, manual backwards stepwise selection was used to eliminate variables that did not contribute significantly to the model using a significance threshold typical for prognostic model research of p=0.20.[21] Clinically relevant variables determined a priori including HbA1c, sex and age were forced to remain in the model regardless of the p value.

From here, two separate sex-specific models were developed. The model for females considered all of the potential predictor variables available for at least 66.6% of the study cohort. The model for males did not include polycystic ovarian syndrome or gestational diabetes as potential predictor variables. The following steps were followed separately for the male and female models: (1) flexible parametric modelling was used to fit the final prognostic model and Rubin's rules were applied to combine the results across the imputed datasets; (2) the linear predictor was calculated for each patient; (3) Harrell's C statistics, Somers' D statistics and calibration slopes were calculated for each imputed dataset and averaged.[22]

## Validation of the models

The models were internally validated to correct for overfitting. Internal validation was performed separately for the male and female models. The same methodology

used for multiple imputation in the development dataset was used for the validation dataset. Internal validation was performed as described by Harrell *et al* and Snee.[23 24] The developed model was applied to the validation dataset and the performance was quantified.[23] A global shrinkage factor (the mean calibration slope) was applied to the beta coefficients from the developed model. The restricted cubic splines and constant relating to the baseline of the model were re-estimated to maintain overall calibration.[25]

Four risk groups (high, medium high, medium low and low) were defined by the 15th, 50th and 85th percentiles of the linear predictor (the model's prognostic index distribution). A Kaplan-Meier curve was plotted for all four groups. Discrimination was visualised by the difference in observed type 2 diabetes-free probability among the groups.

To evaluate the calibration, each imputed dataset was divided into deciles based on the linear predictor of type 2 diabetes risk. The predicted probability of developing type 2 diabetes (x-axis) and the observed fraction that developed type 2 diabetes at 10 years (y-axis) were plotted for each decile risk group. The slope of this line is the calibration slope; a reference line showing perfect calibration was also plotted.

All analyses were performed in Stata V.15 and SAS V.9.4; nominal statistical significance was defined at $p < 0.05$.

### Patient and public involvement
Members of the public were involved in the priority-setting and question-development stages of this study.

## RESULTS
### Study population
A total of 289 754 adult patients were identified from CPRD with an HbA1c test result indicating NDH on or before 31 December 2017. Patients were excluded if they had pre-existing Type 2 diabetes (n=58 296) or type 1 diabetes (n=822). Patients with one or more prescriptions for metformin within 6 months before the index date were also excluded (n=10 260). Patients were further excluded if the first recorded test indicating NDH occurred before the start of the study on 1 January 2000 (n=65 370), or if the date of death preceded the date of the first recorded test indicating NDH (n=301) as these data were likely misreported. There were 154 705 patients that met the inclusion criteria and were included in the cohort (see online supplemental figure 1); 109 077 patients were included in the development dataset (50 049 males and 59 028 females) and 45 628 patients in the validation dataset (21 014 males and 24 614 females).

In the development dataset, there were 9332 patients, including 4719 males and 4613 females, diagnosed with type 2 diabetes during a total of 293 238 person-years of follow-up. The mean follow-up for the development dataset was 2.7 years (SD 2.4, range 0–10 years). In the validation dataset, there were 3783 patients, including 1893 males and 1890 females, diagnosed with type 2 diabetes

during a total of 115 113 person-years of follow-up. The mean follow-up for the validation dataset was 2.5 years (SD 2.3, range 0–10 years).

### Baseline characteristics
Table 2 shows the baseline characteristics of patients in the development and validation datasets and for patients with no missing data. The distributions of continuous variables in the development and validation datasets are shown in (see online supplemental figure 2).

The development dataset included 54.1% female and 12.9% non-white ethnicity; corresponding values in the validation dataset were 53.9% and 11.2%. Within the development dataset, 20.5% of patients were current alcohol users and 13.6% were current smokers compared with 20.6% and 13.7%, respectively, within the validation dataset. The percentage of patients with prescriptions of each medication was similar between the development and validation datasets. The most commonly prescribed medication was antihypertensives (58.0% in the development and 58.5% in the validation dataset), while the least common was atypical antipsychotics (2.6% and 2.4%, respectively). Of the 38 918 patients prescribed corticosteroids in the development dataset, 10 711 (27.5%) were prescribed oral medication, 19 192 were non-oral (49.3%) and 9015 were prescribed both (23.2%; data not shown). For the validation dataset, there were 16 172 patients prescribed corticosteroids including 4637 (28.7%) oral, 7781 (48.1%) non-oral and 3754 prescribed both (23.2%). The medical/family history was similar between the development and validation datasets. The most common medical/family history condition was depression (27.2% in the development and 27.9% in the validation dataset), while the least common was a family history of diabetes (0.1% in both datasets). The mean HbA1c at the index date was the same for development and validation patients, 43.5 mmol/mol (SD 1.2) or 6.1% (0.1%). Further, observed cholesterol and blood pressure were similar between the development and validation datasets.

### Incidence rates of type 2 diabetes
Online supplemental table 3 shows the incidence of type 2 diabetes in total and in the development and validation datasets. The total incidence of type 2 diabetes was 32.1 (95% CI 31.6 to 32.7) per 1000 person-years (py): 31.8 (95% CI 31.2 to 32.5) in the development and 32.9 (95% CI 31.8 to 33.9) in the validation dataset. The largest rate difference between the development and validation datasets was for patients with a history of learning disability; the rate was 30.0 (95% CI 21.1 to 42.7) per 1000 py in the development dataset compared with 41.2 (95% CI 27.6 to 61.5) in the validation dataset.

### Predictor variables
Variables missing for more than 33.3% of the study cohort were eliminated as potential predictor variables including waist circumference (missing for 99.3% of patients), liver

**Table 2** Characteristics of cohort at the index date in total, by number of missing variables and by dataset

| | | | Missing variables | | Dataset | |
| --- | --- | --- | --- | --- | --- | --- |
| | | Total | One or more | None | Development | Validation |
| Total | | n=154705 | n=91409 | n=63296 | n=109077 | n=45628 |
| Age (years) | | 64.9 (14.2) | 64.2 (14.9) | 65.9 (13.1) | 64.8 (14.2) | 65.0 (14.2) |
| Sex | Male | 71063 (45.9%) | 40518 (44.3%) | 30545 (48.3%) | 50049 (45.9%) | 21014 (46.1%) |
| | Female | 83642 (54.1%) | 50891 (55.7%) | 32751 (51.7%) | 59028 (54.1%) | 24614 (53.9%) |
| Ethnicity | Non-white | 14116 (12.4%) | 6683 (13.3%) | 7433 (11.7%) | 10239 (12.9%) | 3877 (11.2%) |
| | White | 99468 (87.6%) | 43605 (86.7%) | 55863 (88.3%) | 68870 (87.1%) | 30598 (88.8%) |
| | Unknown | 41121 | 41121 | 0 | 29968 | 11153 |
| Current alcohol user | | 31722 (20.5%) | 14867 (16.3%) | 16855 (26.6%) | 22320 (20.5%) | 9402 (20.6%) |
| Current smoker | | 21126 (13.7%) | 11677 (12.8%) | 9449 (14.9%) | 14861 (13.6%) | 6265 (13.7%) |
| Medication | Antihypertensives | 90005 (58.2%) | 47424 (51.9%) | 42581 (67.3%) | 63290 (58.0%) | 26715 (58.5%) |
| | Atypical antipsychotics | 3959 (2.6%) | 2541 (2.8%) | 1418 (2.2%) | 2845 (2.6%) | 1114 (2.4%) |
| | Aspirin | 41986 (27.1%) | 22404 (24.5%) | 19582 (30.9%) | 29726 (27.3%) | 12260 (26.9%) |
| | Corticosteroids | 55090 (35.6%) | 33167 (36.3%) | 21923 (34.6%) | 38918 (35.7%) | 16172 (35.4%) |
| | Statins | 74166 (47.9%) | 39425 (43.1%) | 34741 (54.9%) | 52393 (48.0%) | 21773 (47.7%) |
| Medical/family history | Schizophrenia/bipolar | 2093 (1.4%) | 1189 (1.3%) | 904 (1.4%) | 1493 (1.4%) | 600 (1.3%) |
| | Cardiovascular disease | 18483 (11.9%) | 9608 (10.5%) | 8875 (14.0%) | 12862 (11.8%) | 5621 (12.3%) |
| | Depression | 42364 (27.4%) | 26066 (28.5%) | 16298 (25.7%) | 29627 (27.2%) | 12737 (27.9%) |
| | Learning disability | 744 (0.5%) | 446 (0.5%) | 298 (0.5%) | 478 (0.4%) | 266 (0.6%) |
| | Diabetes in family | 195 (0.1%) | 117 (0.1%) | 78 (0.1%) | 159 (0.1%) | 36 (0.1%) |
| | PCOS | 840 (0.5%) | 595 (0.7%) | 245 (0.4%) | 576 (0.5%) | 264 (0.6%) |
| | Gestational diabetes | 762 (0.5%) | 592 (0.6%) | 170 (0.3%) | 567 (0.5%) | 195 (0.4%) |
| | Renal/kidney disease | 17126 (11.1%) | 9109 (10.0%) | 8017 (12.7%) | 11810 (10.8%) | 5316 (11.7%) |
| | Sleep apnoea | 2289 (1.5%) | 1317 (1.4%) | 972 (1.5%) | 1594 (1.5%) | 695 (1.5%) |
| Clinical measures | HbA1c (mmol/mol) | 43.5 (1.5) | 43.5 (1.5) | 43.5 (1.5) | 43.5 (1.5) | 43.5 (1.5) |
| | Cholesterol (mmol/L) | 5.2 (1.2) | 5.3 (1.2) | 5.2 (1.2) | 5.2 (1.2) | 5.2 (1.2) |
| | Systolic BP (mmHg) | 138.1 (18.5) | 137.8 (18.8) | 138.2 (18.4) | 138.0 (18.6) | 138.2 (18.5) |
| | Diastolic BP (mmHg) | 80.0 (11.0) | 79.6 (11.0) | 80.2 (11.0) | 79.9 (11.0) | 80.1 (10.9) |

Continuous variables are given as the mean (SD). Categorical variables are given as the number (%). Index of multiple deprivation, BMI, pulse, liver function test and waist circumference are not included in the table since these measures are not available for >33.3% of the cohort.
BMI, body mass index; BP, blood pressure; PCOS, polycystic ovarian syndrome.

function test (99.2% missing), pulse rate (86.5% missing), BMI (73.6% missing), and deprivation (41.1% missing).

For flexible parametric modelling, 3 df were selected for the restricted cubic spline function used for the baseline hazard (AIC=81482, BIC=81520). This places two knots at percentile positions 33 and 67 of the distribution of the uncensored log survival times. Linear was the best fit for all continuous potential predictor variables; no fractional polynomial transformations were selected. Imputation did not significantly alter the distribution of cholesterol, blood pressure and ethnicity (see online supplemental table 4).

The following potential predictor variables were removed during the backwards selection process: atypical antipsychotics, cholesterol, history of a learning disability, a history of depression, a history of schizophrenia or bipolar affective disorder and ethnicity. The final male model comprised 14 predictor variables including HbA1c, systolic blood pressure, diastolic blood pressure, age, smoking, alcohol use; prescribed medications: antihypertensives, aspirin, corticosteroids, statins and medical history of: cardiovascular disease, renal/kidney disease, sleep apnoea; and family history of diabetes (table 3). The female model included two additional predictors, medical history of polycystic ovarian syndrome and gestational diabetes (table 3).

**Table 3** Development and final coefficients for the male and female prognostic models

| Predictor | Male | | | | Female | | | |
|---|---|---|---|---|---|---|---|---|
| | Development model | | P value | Final model | Development model | | P value | Final model |
| | Coefficient | 95% CI | | Coefficient | Coefficient | 95% CI | | Coefficient |
| HbA1c (mmol/mol) | 0.35048 | 0.33231–0.36866 | 0.000 | 0.34124 | 0.38494 | 0.36673–0.40315 | 0.000 | 0.38255 |
| Age | −0.00310 | −0.00579 – −0.00040 | 0.024 | −0.00302 | −0.00465 | −0.00737 – −0.00193 | 0.001 | −0.00462 |
| Current alcohol user | 0.05866 | −0.00659–0.12391 | 0.078 | 0.05711 | 0.03588 | −0.03874–0.11050 | 0.346 | 0.03566 |
| Current smoker | −0.13053 | −0.21393 – −0.04714 | 0.002 | −0.12709 | −0.11355 | −0.20407 – −0.02302 | 0.014 | −0.11284 |
| Antihypertensive | 0.13787 | −0.03490–0.31064 | 0.118 | 0.13423 | 0.23830 | −0.01509–0.49169 | 0.065 | 0.23682 |
| Aspirin | 0.10917 | 0.04131–0.17703 | 0.002 | 0.10629 | 0.13078 | 0.06142–0.20015 | 0.000 | 0.12997 |
| Corticosteroids | 0.13683 | 0.07441–0.19926 | 0.000 | 0.13322 | 0.12593 | 0.05951–0.19234 | 0.000 | 0.12515 |
| Statins | 0.65113 | 0.58046–0.72180 | 0.000 | 0.63396 | 0.66886 | 0.60170–0.73603 | 0.000 | 0.66471 |
| Cardiovascular disease | −0.08578 | −0.16955 – −0.00201 | 0.045 | −0.08352 | −0.11919 | −0.22249 – −0.01590 | 0.024 | −0.11845 |
| Diabetes in family | 0.65379 | 0.10842–1.19917 | 0.019 | 0.63655 | 0.37641 | −0.31827–1.07110 | 0.288 | 0.37408 |
| Polycystic ovarian syndrome | – | – | – | – | 0.22766 | −0.08223–0.53755 | 0.150 | 0.22625 |
| Gestational diabetes | – | – | – | – | 0.49865 | 0.24068–0.75661 | 0.000 | 0.49555 |
| Renal/kidney disease | −0.05138 | −0.15758–0.05481 | 0.343 | −0.05003 | −0.13741 | −0.23253 – −0.04229 | 0.005 | −0.13655 |
| Sleep apnoea | 0.08901 | −0.09730–0.27532 | 0.349 | 0.08666 | 0.35832 | 0.04615–0.67048 | 0.024 | 0.35609 |
| Systolic blood pressure (mm Hg) | 0.00594 | 0.00383–0.00805 | 0.000 | 0.00578 | 0.00599 | 0.00347–0.00852 | 0.000 | 0.00596 |
| Diastolic blood pressure (mm Hg) | 0.00359 | 0.00009–0.00708 | 0.044 | 0.00349 | 0.00053 | −0.00333–0.00439 | 0.784 | 0.00053 |
| Restricted cubic spline 1 | 0.96661 | 0.94161–0.99160 | 0.000 | 0.96661 | 0.93046 | 0.90612–0.95481 | 0.000 | 0.93046 |
| Restricted cubic spline 2 | −0.03565 | −0.05114 – −0.02016 | 0.000 | −0.03565 | −0.02957 | −0.04468 – 0.01445 | 0.000 | −0.02957 |
| Restricted cubic spline 3 | 0.03708 | 0.02516–0.04901 | 0.000 | 0.03708 | 0.01933 | 0.00740–0.03127 | 0.002 | 0.01933 |
| Constant | −19.55409 | −20.40687 – −18.70131 | 0.000 | −19.55409 | −20.84774 | −21.70300 – −19.99247 | 0.000 | −20.84774 |

Final model coefficients include adjustment for over-fitting.

## Calibration

Using the developed model, (see online supplemental figure 3) shows an example of the calibration between expected and observed probabilities of developing type 2 diabetes at 10 years of follow-up within one of the imputed female and male validation datasets. There were slight differences between plots from the different imputed datasets due to the different values imputed for predictors. Using Rubin's rules to combine the results across imputed datasets, the calibration slope was 0.974 (95% CI 0.905 to 1.042) for males and 0.994 (95% CI 0.931 to 1.057) for females. This indicates that the developed models were slightly overfitted. A uniform shrinkage factor (S=0.974 for males and S=0.994 for females) was applied to each developed model's beta coefficients before recalibrating the baseline function of the final model.

## Discrimination

There was relatively good separation, or discrimination, between risk groups for both males and females when the developed models were fitted using the validation dataset. (See online supplemental figure 4) shows an example using one of the imputed validation datasets. There were slight differences between plots from the different imputed datasets due to the different values imputed for predictors. For both males and females, the log-rank test for all imputed datasets indicated that the survivor functions were different between risk groups (p<0.001 for both males and females). Furthermore, validation showed that the male model discriminated reasonably well with mean Harrell's C statistic across imputed datasets of 0.701 and Somers' D statistic of 0.402; for the female model, the corresponding statistics were 0.718 and 0.436 (table 4). These values suggest slightly better discrimination for the female model.

## DISCUSSION

Although several prognostic and diagnostic models for predicting type 2 diabetes-related outcomes have been developed and validated within the UK, none to date has been specifically developed in a population with NDH, for whom the risk profile is likely different than the general population. The available evidence shows that the incidence of type 2 diabetes in the cohort of patients used to develop the QDiabetes-2018 risk assessment tool was 4.17 (95% CI 4.15 to 4.19) per 1000 person-years.[17] Those included in our study were significantly more likely to develop type 2 diabetes. In fact, the incidence in our

development cohort was nearly eight times that of the QDiabetes-2018 development cohort. Therefore, we have developed and validated pragmatic sex-specific prognostic models for predicting the risk of developing type 2 diabetes in those with NDH, which could be used for targeting referral to the NHS DPP. Our models include important risk factors for people that already have NDH.

Since the primary aim of this study was to develop models that could be easily implemented using routinely collected data, in the variable selection process we closely considered data availability and excluded variables with high levels of missing data, including waist circumference, liver function, pulse rate, BMI and deprivation. Waist circumference and BMI are key risk factors for type 2 diabetes, but these measures may not be obtained due to lack of time and other practical or perceived barriers.[25] BMI, in particular, has been included in many existing type 2 diabetes models.[10] However, the inclusion of BMI must be balanced with practicality, given that our data showed BMI (or height and weight) were infrequently recorded in a primary care setting.

Since the models were developed using observational primary care data, the accuracy of coding, particularly of the outcome, has the potential to affect model development. Research published in 2011 found that miscoding, misdiagnosis and misclassification of diabetes was common in UK primary care.[26] However, in more recent years, implementation of the UK Quality and Outcomes Framework has resulted in better coding of type 2 diabetes, specifically within CPRD.[27 28] With improved interoperability, the launch of Systematized Nomenclature of Medicine (SNOMED) is expected to further boost coding accuracy.[29] Since this research used data initially recorded for managing the care of individual patients, there are also a number of potential sources of bias. To address this, the study cohort included only patients that are considered by CPRD of acceptable research standards. Further, clinical measures that were not biologically plausible and likely misreported were excluded. In most cases, another value that was biologically plausible was available within the same period for the patient.

This study has several strengths. These models are for use in primary care. Therefore, we used a primary care database (CPRD) to develop the models. In recent years the HbA1c assay has been the preferred method to diagnose NDH and type 2 diabetes compared with oral glucose tolerance or fasting plasma glucose tests.[30] Therefore, these models were developed using HbA1c to

**Table 4** Male and female prognostic model mean performance statistics across imputed datasets

| Measure | Male | | Female | |
|---|---|---|---|---|
| | Development | Validation | Development | Validation |
| Harrell's C | 0.700 | 0.701 | 0.720 | 0.718 |
| Somers' D | 0.401 | 0.402 | 0.441 | 0.436 |
| Calibration slope | 1.000 | 0.974 | 1.000 | 0.994 |

quantify blood glucose. The large sample size allowed for a sufficient number of events per predictor parameter. We considered a range of predictors specifically selected due to clinical relevance to development of type 2 diabetes. Continuous predictors were not categorised, so there was no loss of information. The decision to develop sex-specific models was based on the presence of some sex-specific risk factors, like history of gestational diabetes. Additionally, we identified new risk factors not included in the 2018 update of QDiabetes, which was developed within the general population.[17] These risk factors include history of sleep apnoea, blood pressure, alcohol use, prescription of antihypertensives and prescription of aspirin.

This study also had several limitations. The primary limitation is the splitting of the cohort into development and validation datasets instead of using a fully external database to validate the model. However, given the size of the cohort and the large number of events, this likely had little effect on model development. Furthermore, to ensure case mix, non-random selection was used to split the cohort. The outcome for this study was defined using a single medcode or test result indicating type 2 diabetes. In practice, this would typically be confirmed via a follow-up test. Another limitation is that the models included predictor variables obtained at one point in time including a single HbA1c measure to determine NDH. However, the models could be adjusted to include time-varying predictors relatively easily. Methods such as land marking or joint models could be used to model changes in predictors over time. Some predictor variables were self-reported including smoking, alcohol use and family history of diabetes. The proportion of non-current smokers is in line with a similar study while the proportion of patients with a family history of diabetes in this study was much lower than that reported in a similar study.[17] This may indicate that family history of diabetes is not established in clinical practice or established but not recorded within the CPRD. Prescriptions issued were used as a proxy for current medication. Patients may not have filled the prescription or adhered to the medication. Because this was an open cohort and the number of people diagnosed with NDH has increased in recent years, the mean follow-up time was short—2.7 years for patients in the development dataset and 2.5 years for patients in the validation dataset. However, 14896 patients in the development dataset and 5678 patients in the validation dataset had five or more years of follow-up. Therefore, based on existing research, we believe that there was sufficient follow-up time to determine risk for progression to type 2 diabetes. HES and ONS linkage was only available for 59.0% of patients in the cohort. If linkage to ONS was not available and a date of death was provided in CPRD, then the CPRD date was used. While ONS is the gold standard for date of death, deaths are less well coded in CPRD. It is possible that deaths for some patients without linkage to ONS were never coded in CPRD, and the patients were not censored accordingly. However, this likely only affected a few patients. It is possible that patients receiving non-metformin oral hypoglycaemic agents at baseline were included in the cohort. However, it is highly unlikely that a patient would have been prescribed a non-metformin oral hypoglycaemic agent without meeting any of the other exclusion criteria. Finally, there may have been additional predictor variables that were not considered either because they are not collected as part of routine clinical care or because they are not among the known traditional risk factors for type 2 diabetes.

Similar to the QRISK cardiovascular disease risk algorithm, the models presented are designed to be integrated into primary care computer systems to automatically calculate risk.[31] At the time of the first HbA1c test indicating NDH, a risk score could be automatically generated using the HbA1c measure along with clinical, prescription and diagnoses data already contained in the individual's electronic health record. Additionally, the algorithm for imputing missing data could also be implemented automatically. Rather than referring all adults with NDH to the NHS DPP, healthcare providers could prioritise referrals for people at high risk for progressing to type 2 diabetes.

The NHS DPP is a limited resource and does not have current capacity to accommodate all adults with NDH in England. People are referred to the NHS DPP through the NHS Health Check programme, aimed at people aged 40–74, or people with NDH identified through opportunistic assessment or as part of routine clinical care.[9] Eligibility for the NHS DPP is typically determined through an HbA1c measure or, less frequently, an oral glucose tolerance test. However, this study has identified additional factors to stratify further the risk of developing type 2 diabetes within this high-risk group. Targeting referrals may be a more cost-effective and efficient way to deliver the NHS DPP. The male and female prognostic models we developed and validated could be used to identify and target those most at risk of developing type 2 diabetes for referral to the NHS DPP. Implementation of these models would standardise the NHS DPP identification and referral process to be consistent across sites and based on information already collected as part of primary care. The next step is to determine the optimum risk threshold to accurately identify patients that will develop type 2 diabetes.

**Acknowledgements** This study is based on data from the Clinical Practice Research Datalink GOLD database obtained under licence from the UK Medicines and Healthcare products Regulatory Agency. However, the interpretation and conclusions contained in this article are those of the authors alone. The authors gratefully acknowledge Leicester Real-World Evidence Unit (LRWE) for facilitating the download of CPRD data. LRWE is funded by University of Leicester, National Institute for Health Research (NIHR) Applied Research Collaboration East Midlands (ARC EM) and NIHR Leicester Biomedical Research Centre. The interpretation and conclusions contained in this report/article do not necessarily reflect those of the LRWE.

**Funding** This research was supported by a grant from National Institute for Health Research (NIHR) Collaboration for Leadership in Applied Health Research and Care (CLAHRC) East Midlands. This research was also funded in part by the NIHR Leicester Biomedical Research Centre and NIHR Applied Research Collaboration.

**Disclaimer** The funding body had no role in the study design, data collection, analysis, or interpretation; in the writing of the manuscript; or in the decision to submit the manuscript. The views expressed are those of the author(s) and not

necessarily those of the NHS, the NIHR or the Department of Health and Social Care.

**Competing interests**   BC, LG, FZ and SB: none. MJD has acted as consultant, advisory board member and speaker for Novo Nordisk, Sanofi-Aventis, Lilly, Merck Sharp & Dohme, Boehringer Ingelheim, AstraZeneca and Janssen, an advisory board member for Servier and as a speaker for Mitsubishi Tanabe Pharma Corporation and Takeda Pharmaceuticals International Inc. She has received grants in support of investigator and investigator initiated trials from Novo Nordisk, Sanofi-Aventis, Lilly, Boehringer Ingelheim and Janssen. She was a member of the NICE public health guideline for prevention of Type 2 diabetes (NICE PH 38). KK has acted as a consultant and speaker for Novartis, Novo Nordisk, Sanofi-Aventis, Lilly and Merck Sharp and Dohme. He has received grants in support of investigator and investigator-initiated trials from Novartis, Novo Nordisk, Sanofi-Aventis, Lilly, Pfizer, Boehringer Ingelheim and Merck Sharp & Dohme. He is a member of the External Reference Group of the NHS DPP and was Chair of the NICE public health guideline for prevention of Type 2 diabetes (NICE PH 38).

**Patient consent for publication**   Not required.

**Ethics approval**   This research was approved by the Independent Scientific Advisory Committee (ISAC) for Medicines and Healthcare products Regulatory Agency Database Research (protocol 18_238).

**Provenance and peer review**   Not commissioned; externally peer reviewed.

**Data availability statement**   No data are available. Patient-level electronic health records obtained from CPRD cannot be shared. However, the authors will share programming code and aggregate statistics if requested. A list of medcodes used to define Type 2 diabetes, pre-existing type 1 diabetes, and medical and family history as well as product codes used to identify current medication is available at github. com/bc188/Prognostic-model-codes.

**ORCID iDs**
Briana Coles http://orcid.org/0000-0003-4228-8818
Kamlesh Khunti http://orcid.org/0000-0003-2343-7099

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
