## [Reviewer comments · BMJ Open]

ARTICLE DETAILS

TITLE (PROVISIONAL)	Prediction of Type 2 diabetes risk in people with non-diabetic hyperglycaemia: model derivation and validation using UK primary care data
AUTHORS	Coles, Briana; Khunti, Kamlesh; Booth, Sarah; Zaccardi, Francesco; Davies, Melanie; Gray, Laura

VERSION 1 – REVIEW

REVIEWER	Omar Yaxmehen Bello-Chavolla National Institute for Geriatrics, Mexico City, Mexico
REVIEW RETURNED	02-Apr-2020

GENERAL COMMENTS	Thank you for allowing me to review this excellent modelling work to improve prediction of diabetes incidence in patients with non-diabetic hyperglycemia in UK primary care facilities. The authors have crafted a methodologically sound and intricate paper that requires only minor adjustments to fully transmit its core messages. I have only a few comments: 1. HbA1c is featured prominently as a predictor, as expected. Nevertheless, the relative contribution of additional covariates must be analyzed given that HbA1c alone could be driving most of the predictive power. Please carry out an analytic approach whereby you compare a model which includes HbA1c compared to one which excludes it; models can be compared in terms of Harrel's c-statistic, Sommer's Dxy and, most prominently, BIC. An alternative for model comparison would be to use the net-reclassification improvement index (doi: 10.1097/EDE.000000000000018).2. Some variables in the final model include terms which do not reach the significance threshold described in the methods. Did you assess whether the model fit improves using Harrel's c-statistic or BIC after removing said variables? Non-contributing variables could reduce predictive power and its removal should be considered for any given model.3. Regarding the imputation procedure, I recognize the relevance of multiple imputation to preserve variability of variables with missing values; however, given the rather large imputation threshold, a comparison of key descriptive measures for imputed variables (mean, standard deviation, skewness and kurtosis) should be shown to demonstrate that the imputation procedure did not significantly modify variable distributions.4. Did the authors consider transforming beta coefficients to a point system? A coefficient-based procedure seems rather unintuitive for a wide application (DOI: 10.1002/sim.1742).5. Finally, it would be relevant to include the accumulated risk thresholds identified in Kaplan-Meier analyses, as a means to translate the identified risk into a useful metrics for clinicians.
--

REVIEWER	Toru Aizawa Aizawa Hospital
REVIEW RETURNED	21-Apr-2020

GENERAL COMMENTS	Coles et al. analyzed a large number of subjects with prediabetes to identify risk factors for conversion to diabetes. The data is potentially important but the study have problems. <Major problems>  1. It is unclear what is a novel finding or findings in this study. See Minor problems #2 below. 2. Entry criteria is basically HbA1c 6.0%-6.4% with or without pharmacological treatment for hyperglycemia. Those receiving metformin at baseline were excluded but those receiving non-metformin oral hypoglycemic agents were included. This means that the study population included patients with diabetes at baseline. 3. Upon follow-up, development of diabetes was diagnosed by A1c greater than 6.4%, random sample glucose greater than 199 mg/dL or FPG greater than 125 mg/dL. Importantly, the latter two criteria were not exclusion criterion of diabetes at baseline. In this reviewer's experience, two to three percent of subjects with A1c 6.0% - 6.4% have FPG greater than 125 mg/dL. Thus, the grouping of the study population is inappropriate. <Minor problems>  1. The study population was rather old: the mean age was 65 years. This makes generalization of the data difficult. 2. When you put "conversion of prediabetes to diabetes" to pubmed, many papers are hit after Ref 13, some of which is highly relevant to their study so that worth to cite and discuss. 3. Observation period was rather short to gain a meaningful predictive data for a disease like diabetes which has a very long time line. 4. For primary care physicians, it is advisable to prepare a simple risk calculator for conversion of NDH to diabetes.
---

REVIEWER	Loai Albarqouni Bond University
REVIEW RETURNED	28-Jun-2020

GENERAL COMMENTS	Thanks for inviting me to review this manuscript. In this article, the authors reported the development and validation of a model to predict the development of type 2 diabetes among non-diabetic hyperglycaemic adults in the UK. Major comments The authors should consider discussing the literature highlighting that the epidemic of 'pre-diabetes' is more of an overdiagnosis or too much medicine – see https://pubmed.ncbi.nlm.nih.gov/25028385/ and https://pubmed.ncbi.nlm.nih.gov/29592924/ Authors should justify developing a risk prediction tool specific for 'prediabetic' individuals to predict 10-yr risk of developing diabetes and how this might be different than QDiabetes-2018 https://www.bmj.com/content/359/bmj.j5019
--

	Authors might consider reporting more details about the trial mentioned in the second paragraph in the introduction (e.g. follow-up time, uncertainty around effect estimates, and baseline population). In brief, my main concern is the added value of this research to clinical practice and research literature.
--	--

VERSION 1 – AUTHOR RESPONSE

Reviewer 1: Omar Yaxmehen Bello-Chavolla

1. HbA1c is featured prominently as a predictor, as expected. Nevertheless, the relative contribution of additional covariates must be analyzed given that HbA1c alone could be driving most of the predictive power. Please carry out an analytic approach whereby you compare a model which includes HbA1c compared to one which excludes it; models can be compared in terms of Harrel's c-statistic, Sommer's Dxy and, most prominently, BIC. An alternative for model comparison would be to use the net-reclassification improvement index (doi: 10.1097/EDE.0000000000000018).

Though research has shown that HbA1c is the primary predictor of development of Type 2 diabetes, using HbA1c as the primary determinant for referral to the Diabetes Prevention Programme (DPP) results in over-referral to the programme which has finite capacity. Therefore, the aim of this study was to develop a prognostic model that incorporates factors in addition to HbA1c to identify the subset of non-diabetic hyperglycaemic patients that are at high risk of developing Type 2 diabetes. These patients should receive priority referral to the DPP. Additionally as the implementation of such models would be through software which would interrogate existing primary care record data (such as the QDiabetes tool) the number of covariates included in the model is not of importance given there is sufficient power to model these robustly in routine databases. Comparative statistics certainly support this approach. Using one of the imputed male datasets as an example, for the full model, the BIC was 15,257, Harrell's C was 0.703, and Somers' D was 0.406. For the model with HbA1c only, these statistics were 15,373, 0.679, and 0.357. Therefore, the full model predicts progressing to Type 2 diabetes better than HbA1c alone.

2. Some variables in the final model include terms which do not reach the significance threshold described in the methods. Did you assess whether the model fit improves using Harrel's c-statistic or BIC after removing said variables? Non-contributing variables could reduce predictive power and its removal should be considered for any given model.

The models were developed using a pre-planned methodology. Degrees of freedom for the baseline hazard function were determined, fractional polynomial transformations were checked, and then backward stepwise selection using the significance threshold of $p=0.20$ was performed on the entire development cohort. Clinically relevant variables determined *a priori* including HbA1c, sex, and age were forced to remain in the model regardless of the p-value. This selection was not stratified by sex. The final selected variables were then used in sex-specific regression to create separate prognostic models for males and females. As a result, some of the p values in the final sex-specific models are above the threshold used for the backward selection procedure.

3. Regarding the imputation procedure, I recognize the relevance of multiple imputation to preserve variability of variables with missing values; however, given the rather large imputation threshold, a comparison of key descriptive measures for imputed variables (mean, standard deviation, skewness and kurtosis) should be shown to demonstrate that the imputation procedure did not significantly modify variable distributions.

The key descriptive measures are now presented as Supplementary Table S4 and the reference to the table has been added to the manuscript page 16. Additionally, we present graphs for the continuous variables below for review. However, since we have a significant number of tables and graphs

accompanying this manuscript, we have opted not to include these within the manuscript materials. However, if the Editor feels these are helpful then we would be happy to include them.

4. Did the authors consider transforming beta coefficients to a point system? A coefficient-based procedure seems rather unintuitive for a wide application (DOI: 10.1002/sim.1742).

Converting beta coefficients to a point system would result in a loss of information. This model is designed to be implemented as an electronic health record-based algorithm and the output, an individualized probability or risk of progressing to Type 2 diabetes, would be calculated automatically based on data already existing within the patient's medical record. Therefore, we are not concerned about intuitiveness of the calculation. The outputted risk would then inform the healthcare provider's decision to refer/not refer the patient to the Diabetes Prevention Programme. A number of risks can already be automatically calculated within a patient's electronic health record including the electronic Frailty Index (eFI), CHA₂DS₂-VASc for atrial fibrillation stroke risk, Time in Therapeutic Range (TTR), and QRISK2 for cardiovascular disease risk.

5. Finally, it would be relevant to include the accumulated risk thresholds identified in Kaplan-Meier analyses, as a means to translate the identified risk into a useful metrics for clinicians.

In an effort not to expand recommendations beyond our research question, we have not prescribed a risk threshold for referral to the Diabetes Prevention Programme (DPP). To demonstrate discrimination, we used the 15th, 50th and 85th percentiles of the linear predictor (prognostic index) for one of the imputed datasets. However, we did not provide the linear predictor values for the Kaplan-Meier as these differ depending on the imputed dataset. Further research needs to be done to identify an appropriate risk threshold, which must be balanced with DPP capacity and funding and modelled cost-effectiveness.

Reviewer 2: Toru Aizawa

<Major problems>

1. It is unclear what is a novel finding or findings in this study. See Minor problems #2 below.

Our study was based on a different population than that used to develop existing risk scores for Type 2 diabetes. Existing risk scores for Type 2 diabetes, QDiabetes-2018 being the most popular, were developed to predict risk in a general population, whereas our model is specific to a population with non-diabetic hyperglycaemia whose risk profile is different than that of the general population. For instance, the incidence of Type 2 diabetes in our development cohort was nearly eight times that of the QDiabetes-2018 development cohort. Therefore, we were able to identify risk factors not included in

QDiabetes-2018, including history of sleep apnoea, blood pressure, alcohol use, prescription of antihypertensives, and prescription of aspirin.

2. Entry criteria is basically HbA1c 6.0%-6.4% with or without pharmacological treatment for hyperglycemia. Those receiving metformin at baseline were excluded but those receiving non-metformin oral hypoglycemic agents were included. This means that the study population included patients with diabetes at baseline.

Guidelines in the UK recommend metformin as first line glucose lowering therapy. Non-metformin oral hypoglycaemic agents may be prescribed to metformin-intolerant patients. However, based on the authors' clinical experiences, it is highly unlikely that a patient would receive a prescription for a non-metformin oral hypoglycaemic agent for Type 2 diabetes without meeting one or more of the exclusion criteria (an HbA1c/random sample glucose/FPG test, preceding metformin prescription, or a diagnosis code for Type 2 diabetes recorded). Though unlikely, we have added this to the limitations section (see page 22).

3. Upon follow-up, development of diabetes was diagnosed by A1c greater than 6.4%, random sample glucose greater than 199 mg/dL or FPG greater than 125 mg/dL. Importantly, the latter two criteria were not exclusion criterion of diabetes at baseline. In this reviewer's experience, two to three percent of subjects with A1c 6.0% - 6.4% have FPG greater than 125 mg/dL. Thus, the grouping of the study population is inappropriate.

Thank you for highlighting this. Upon review of the programming code, we used the same criteria for the outcome and to exclude patients with Type 2 diabetes at baseline. We excluded patients with a diagnosis code for Type 2 diabetes or any of the following before the index date: HbA1c greater than 47 mmol/mol (6.4%), random sample glucose greater than 199 mg/dL, or FPG greater than 125 mg/dL. This has been added to page 7 of the manuscript document.

<Minor problems>

1. The study population was rather old: the mean age was 65 years. This makes generalization of the data difficult.

Though patients are presenting at younger ages with non-diabetic hyperglycaemia, it remains predominantly a disease of the elderly. The CPRD database is broadly representative of the UK general population in terms of age, sex and ethnicity. Since our cohort of patients with non-diabetic hyperglycaemia was obtained from this database, we believe our cohort is representative of the non-diabetic hyperglycaemic population within the UK.

2. When you put "conversion of non-diabetic hyperglycaemia to diabetes" to pubmed, many papers are hit after Ref 13, some of which is highly relevant to their study so that worth to cite and discuss.

Following the reviewer's suggestion, we found two publications. The first (Knowles *et al.*) provides information on the Diabetes Prevention Programme which we deem not strictly related to the goal of our analysis. The second, Oka *et al.*, is focused on progression from normal glucose levels to non-diabetic hyperglycaemia, whereas our study focused on progression from non-diabetic hyperglycaemia to Type 2 diabetes. However, it may be possible that we have missed some studies. We would be grateful to the reviewer to point to other relevant studies we may have missed and could be useful for our discussion.

Knowles S, Cotterill S, Coupe N, Spence M. Referral of patients to diabetes prevention programmes from community campaigns and general practices: mixed-method evaluation using the RE-AIM framework and Normalisation Process Theory. *BMC Health Serv Res.* 2019;19(1):321. Published 2019 May 22. doi:10.1186/s12913-019-4139-5.

Oka R, Yagi K, Hayashi K, et al. The evolution of non-diabetic hyperglycemia: a longitudinal study. *Endocr J.* 2014;61(1):91-99. doi:10.1507/endocrj.ej13-0359.

3. Observation period was rather short to gain a meaningful predictive data for a disease like diabetes which has a very long time line.

The study includes a sizable number of person-years of follow up: 293,238 person-years for development patients and 115,113 person-years for validation patients. Though the mean follow-up time was relatively short (2.7 years for patients in the development dataset and 2.5 years for patients in the validation dataset), 23,336 (21.4%) patients in the development dataset and 9,123 (20.0%) patients in the validation dataset either progressed to Type 2 diabetes or had at least five years of follow up. Therefore, based on existing research, we believe that there was sufficient follow-up time to determine risk for progression to Type 2 diabetes. This is in line with similar studies. For instance, the 2018 update of QDiabetes by *Hippisley-Cox et al.*, the median follow-up in the derivation cohort was 3.90 years (interquartile range 1.54 to 8.50).

Hippisley-Cox J, Coupland C. Development and validation of QDiabetes-2018 risk prediction algorithm to estimate future risk of type 2 diabetes: cohort study. *BMJ*. 2017;359:j5019. Published 2017 Nov 20. doi:10.1136/bmj.j5019.

4. For primary care physicians, it is advisable to prepare a simple risk calculator for conversion of non-diabetic hyperglycaemia to diabetes.

This model is designed to be implemented as an electronic health record-based algorithm and the output, an individualized probability or risk of progressing to Type 2 diabetes, would be calculated automatically based on data already existing within the patient's routine medical record. The primary care physician would not need to calculate the risk manually. A number of risks are available for automatic calculation within a patient's electronic health record including the electronic Frailty Index (eFI), CHA₂DS₂-VASc for atrial fibrillation stroke risk, Time in Therapeutic Range (TTR), and QRISK2 for cardiovascular disease risk.

Reviewer 3: Loai Albarquni

Major comments

1. The authors should consider discussing the literature highlighting that the epidemic of 'pre-diabetes' is more of an overdiagnosis or too much medicine – see <https://eur03.safelinks.protection.outlook.com/?url=https%3A%2F%2Fpubmed.ncbi.nlm.nih.gov%2F25028385%2F&data=02%7C01%7Cbc188%40leicester.ac.uk%7C15c4680e45f24bbd6d5408d82422a9d5%7Caebeed6a31d44b0195ce8274afe853d9%7C0%7C0%7C637299078579018298&data=tbdnvtY70ICPIAbGmKZdLSJFWDq3Ib3mBzhj9tV12A%3D&reserved=0> and <https://eur03.safelinks.protection.outlook.com/?url=https%3A%2F%2Fpubmed.ncbi.nlm.nih.gov%2F29592924%2F&data=02%7C01%7Cbc188%40leicester.ac.uk%7C15c4680e45f24bbd6d5408d82422a9d5%7Caebeed6a31d44b0195ce8274afe853d9%7C0%7C0%7C637299078579018298&data=le0Fn5WyPmcXgLLjUN3cWC5kqjK2GOInedwUp8Cuwkk%3D&reserved=0>

Thank you for this comment. We agree and we have added a statement about overtreatment to page 6 of the manuscript.

2. Authors should justify developing a risk prediction tool specific for 'prediabetic' individuals to predict 10-yr risk of developing diabetes and how this might be different than QDiabetes-2018 <https://eur03.safelinks.protection.outlook.com/?url=https%3A%2F%2Fwww.bmj.com%2Fcontent%2F359%2Fbmj.j5019&data=02%7C01%7Cbc188%40leicester.ac.uk%7C15c4680e45f24bbd6d5408d82422a9d5%7Caebeed6a31d44b0195ce8274afe853d9%7C0%7C0%7C637299078579018298&data=p6gBJ9Wjv7M141TWlTm%2FIBYj7nHKNM1g9ja9ARPq3Qo%3D&reserved=0>

We believe we have addressed this. Within the Introduction we have provided background on the differing risk profiles of people with non-diabetic hyperglycaemia compared to the general population. Additionally, in the discussion, we highlighted how our cohort different from that used to develop the QDiabetes 2018 risk score and cited the study linked above. The primary difference was, as expected,

our cohort had a much higher incidence of Type 2 diabetes. Finally, we identified new risk factors not included in the 2018 update of QDiabetes, including history of sleep apnoea, blood pressure, alcohol use, prescription of antihypertensives, and prescription of aspirin. Some of these risk factors were considered in the QDiabetes model, but were not significant, while others are new risk factors not previously examined.

3. Authors might consider reporting more details about the trial mentioned in the second paragraph in the introduction (e.g. follow-up time, uncertainty around effect estimates, and baseline population).

Thank you. We have added more information on this trial. See second paragraph, page 5.

VERSION 2 – REVIEW

REVIEWER	Omar Yaxmehen Bello-Chavolla Instituto Nacional de Geriatria
REVIEW RETURNED	30-Jul-2020

GENERAL COMMENTS	Authors have adequately addressed all my previous concerns. This is a strong and relevant manuscript and I would like to thank authors for their thorough revision.
---

REVIEWER	Albarqouni, Loai Bond University Faculty of Health Sciences and Medicine
REVIEW RETURNED	19-Aug-2020

GENERAL COMMENTS	Authors have adequately revised the manuscript addressing the reviewers previous comments.
--